# Meningioma Cell Invasion into DuraGen-Derived Dura Mater: A Case Report

**DOI:** 10.3390/medicines9040030

**Published:** 2022-04-11

**Authors:** Ryota Tamura, Yuki Kuranari, Hideki Orikasa, Makoto Katayama

**Affiliations:** 1Department of Neurosurgery, Keio University School of Medicine, Tokyo 160-8582, Japan; 2Department of Neurosurgery, Kawasaki Municipal Hospital, Kawasaki-ku 210-0013, Japan; e073022fplanet16@gmail.com (Y.K.); makoto.katayama@gmail.com (M.K.); 3Department of Pathology, Kawasaki Municipal Hospital, Kawasaki-ku 210-0013, Japan; orihhym@js6.so-net.ne.jp

**Keywords:** meningioma, invasion, dura mater, DuraGen, artificial, autologous

## Abstract

**Background:** Dura mater infiltration is the main growth pattern of meningiomas. Local recurrence may occur in any type of meningioma, but it is more likely so in atypical meningiomas. Therefore, a wide resection of tumor cell-invaded dura mater is necessary to avoid recurrence. DuraGen^®^ (an artificial dural substitute) can be used for dural reconstruction in meningiomas. Here, we report a rare case of a patient with atypical meningioma that invaded into the DuraGen^®^-derived mature dura mater. **Case presentation:** A 66-year-old female showed a three-time recurrence of atypical meningioma. Simpson grade I resection (en bloc tumor with autologous dura mater and DuraGen^®^-derived dura mater resection) was achieved at the 3rd recurrence. Collagen fibers running regularly and transversely were observed in the DuraGen^®^-derived dura mater resembling the autologous meningeal layer. Meningioma cell invasion, displayed by occasional EMA immunostaining, was observed in the DuraGen^®^-derived dura mater. **Conclusions:** This case indicates that meningioma cells may invade and survive in the DuraGen^®-^derived dura mater. Whether or not DuraGen^®^ is not appropriate as a dural substitute remains unanswered. Further experiences are needed to validate these findings in large sample sizes.

## 1. Introduction

Meningioma is the most common benign intracranial tumor. Meningiomas often invade the dura mater because they originate from arachnoid cap cells beneath the adjacent dura mater [1]. Recurrence frequently occurs at the edge of the dural resection site [2]. Local recurrence occurs in 30.3% at 3 years in patients with gross totally resected atypical meningiomas [2]. Therefore, a wide resection of dura mater, which includes the invading tumor cells, is necessary to avoid recurrence [1,3,4].

Dural reconstruction is required to avoid cerebrospinal fluid (CSF) leakage after the removal of meningiomas. Current surgical techniques of dural repair consist of the use of muscular fascia, periosteum, and artificial dural substitutes. DuraGen^®^ (Integra Life Sciences Corp., Princeton, NJ, USA) is a safe and effective type I collagen matrix graft, manufactured from bovine Achilles tendon [5,6]. At present, DuraGen^®^ can also be used for dural reconstruction in meningiomas [7,8].

Past studies have focused on the association between tumor recurrence and reconstruction tissue, but none have examined the invasion of meningioma cells into artificial dural substitutes [9,10]. Whereas there are numerous reports related to DuraGen^®^ usage and CSF leakage, the literature is sparce, if any, on attempts to relate the tumor biology and the histopathology of the neoplasm to DuraGen^®^ [11,12]. We report herein a rare case of an atypical meningioma that invaded the DuraGen^®^-derived dura mater.

## 2. Case Presentation

A 66-year-old female presented with a headache. Computed tomography (CT) showed a left tentorial meningioma with surrounding brain edema. Simpson grade II resection (macroscopically complete removal of the tumor and its visible extensions with coagulation of its dural attachment) was performed. Four years later, another Simpson grade II resection was performed for a recurrent lesion. We decided not to attempt Simpson 1 resection while the transverse sinus was patent during the first two surgeries. Both were diagnosed as a WHO grade I meningioma. Duraplasty was performed with a Gore-Tex^®^ dura substitute (W.L. Gore & Associates Inc., Phoenix, AZ, USA) for the 1st and 2nd operations. Magnetic resonance imaging (MRI) showed tumor regrowth nine years after the 2nd operation (Figure 1). Simpson grade I resection (en bloc tumor and tentorium resection) was achieved for this 2nd recurrent lesion. The transverse sinus was completely occluded by the tumor and was removed. DuraGen^®^ was placed in a multilayered fashion in order to achieve duraplasty [13]. Fibrin glue was applied to the repaired dura to prevent CSF leakage and migration of the grafts. Absence of CSF leakage was confirmed with a Valsalva maneuver after completion of the dural reconstruction. Postoperative care was as per the standard protocols. A diagnosis of WHO grade II meningioma was made. Radiotherapy was recommended to the patient, although it was refused.

MRI showed rapid tumor regrowth one year after the 3rd operation (Figure 1). The musculocutaneous flap was re-opened. A secondary dura-like bed was observed between the musculocutaneous flap and brain. No CSF leakage was observed post procedure. A thick, fibrous layer resembling an autologous dura mater was observed at the DuraGen^®^ implanted site (Figure 2). Simpson grade I resection (en bloc tumor of the autologous dura mater and DuraGen^®^-derived dura mater resection) was achieved again for the 3rd recurrent lesion (Figure 2). The tumor originated from the edge of the tentorium. The histopathological diagnosis was WHO grade II meningioma, again.

Macroscopic images of the formalin fixed resected specimen are shown in Figure 3A. Microscopic lower magnification demonstrated increased cellular infiltration and frequent fibrous adhesions or connective tissue of the DuraGen^®^-derived dura mater (Figure 3B). The meningeal and periosteal layer is observed in the autologous dura mater. Collagen fibers running regularly and transversely were observed in the DuraGen^®^-derived dura mater resembling an autologous meningeal layer. Neovascularization was also evident in the DuraGen^®^-derived dura mater. Meningioma cell invasion, which was detected by EMA and Ki-67 immunostainings, were partially observed in the DuraGen^®^-derived dura mater (Figure 3B). Currently, the most commonly used immunohistochemistry marker for the diagnosis of meningiomas is EMA [14,15]. The previous study demonstrated that Ki-67 labeling index is the only independent predictor of both tumor recurrence and overall survival in meningioma. It is highly recommended that Ki-67 expression profile is assessed in meningiomas to predict survival [16,17]. The Ki-67 labeling index of the 3rd recurrent lesion was higher than those of the previous lesions (initial lesion: 7%; 1st recurrent lesion: 5%; 2nd recurrent lesion: 8%; 3rd recurrent lesion: 32%).

## 3. Discussion

DuraGen^®^ is frequently used for a variety of conditions and procedures including traumatic brain injury and stroke [18]. DuraGen^®^ can also be used for duraplasty after tumor removal [18]. Whereas the focus of most of these reports are the incidence of CSF leakage following non-watertight reconstruction, none have demonstrated the association between tumor cell invasion and DuraGen^®^-derived dura mater.

Fibroblasts begin to migrate into the matrix 2 to 3 days after implantation of DuraGen^®^. A neodural membrane forms between the dural margins to permanently close the dural defect within 2 weeks. After 6–8 weeks, the implant is resorbed and replaced by dural tissue. After 1 year, the neodura is completely replaced by a mature dura; this can be observed in the present case [5,19]. Previous studies demonstrate that the periosteal layer of the autologous dura mater consists of microfibrils, elastic fibers and extensive amounts of collagen fibers that and form bladed bundles [20,21]. In contrast, the meningeal layer contains more fibroblasts and less extra cellular matrix. An electron microscopy study demonstrates that collagen bundles run in multiplex directions in the periosteal layer, whereas they run regularly and transversely in the meningeal layer [20,21]. DuraGen^®^-derived dura mater resembles the autologous meningeal layer very closely.

Meningiomas have been shown to harbor cancer stem cells, highly resilient cancer cells that employ deregulated stem cell expression profiles and can be the cause of recurrence [22]. Furthermore, meningioma stroma mesenchymal stem cells (MSCs) exist in meningioma specimens, which could be a component of the meningioma cellular microenvironment [23]. DuraGen^®^ is compatible with the culture of various types of stem cells. DuraGen^®^ can be used as a biological membrane to wrap the MSCs and to allow the MSCs to survive in the body for longer periods of time. Packing of MSCs with DuraGen^®^ reduces apoptosis [24]. Meningioma cells are known to invade into different extracranial structures (e.g., muscle tissue and galea) [25]. Meningioma cells may invade and survive in the DuraGen^®^-derived dura mater. Whether or not DuraGen^®^ is not appropriate as a dural substitute remains unanswered.

The main limitation of this report was the small number of cases assessed. Future studies analyzing a large number of patients are warranted to confirm the findings in this study. Aquaporin-1 (AQP-1) and matrix metalloproteinase-1 (MMP-1) are associated with higher recurrence rate [10] and dural invasion [2]. The relationship between AQP-1, MMP-1 and meningioma cell invasion in the DuraGen^®^-derived dura mater will be evaluated in the future study using a large number of patients.

## 4. Conclusions

This is the first case report demonstrating meningioma cell invasion into the DuraGen^®^-derived dura mater. Meningioma cells may invade and survive in the DuraGen^®^-derived dura mater. Whether or not DuraGen^®^ is not appropriate as a dural substitute remains unanswered. Further experiences are needed to validate these findings in large sample sizes.

## Figures and Tables

**Figure 1 medicines-09-00030-f001:**
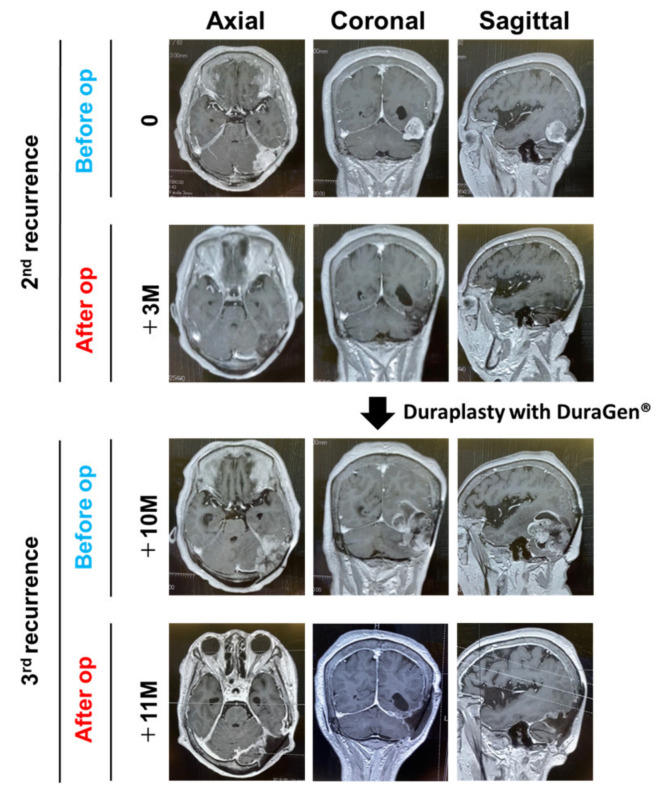
Radiographical images. Preoperative and postoperative gadolinium-enhanced T1-weighted imaging at the 2nd and 3rd recurrences are shown (axial, coronal, and sagittal views). DuraGen^®^ was placed to achieve duraplasty at the operation for the 2nd recurrence. op, operation.

**Figure 2 medicines-09-00030-f002:**
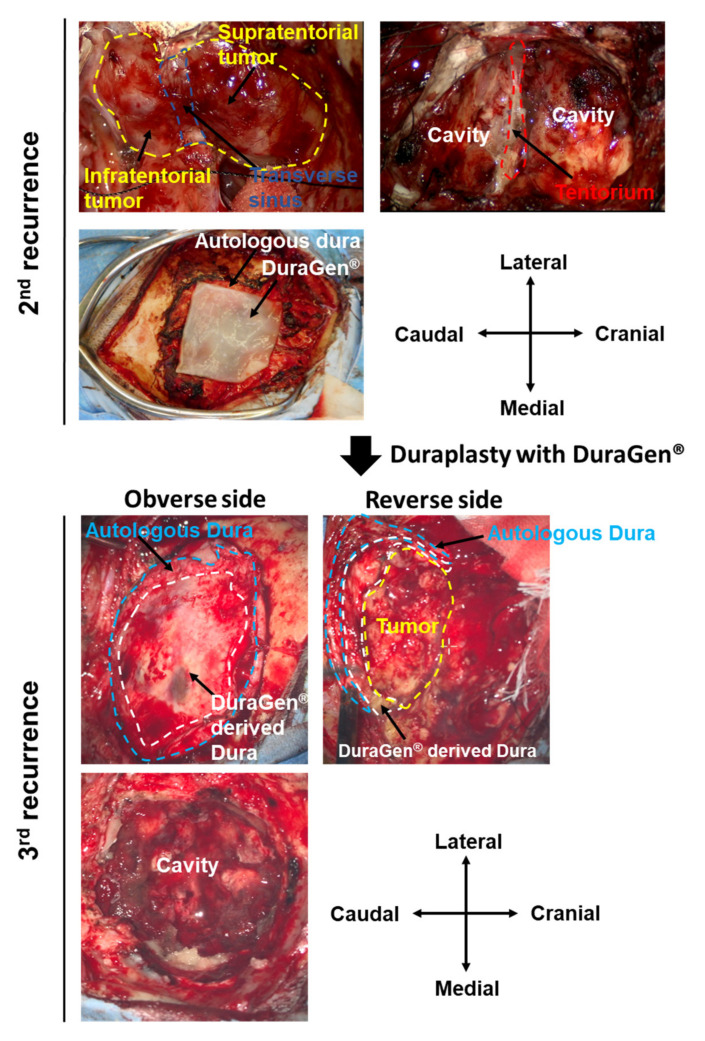
Intraoperative image. Simpson grade I resection (en bloc tumor and tentorium resection) could be achieved for the 2nd recurrent lesion. Transverse sinus occluded by the tumor invasion was completely removed. DuraGen^®^ was placed in order to achieve duraplasty. At the operation for the 3rd recurrence, the musculocutaneous flap was re-opened. Complete neoduralization of the DuraGen^®^ was observed above the tumor. Simpson grade I resection (en bloc tumor, and autologous dura mater and DuraGen^®^-derived dura mater resection) was achieved again for the 3rd recurrent lesion. **Upper panel**: 2nd recurrence; **lower panel**: 3rd recurrence.

**Figure 3 medicines-09-00030-f003:**
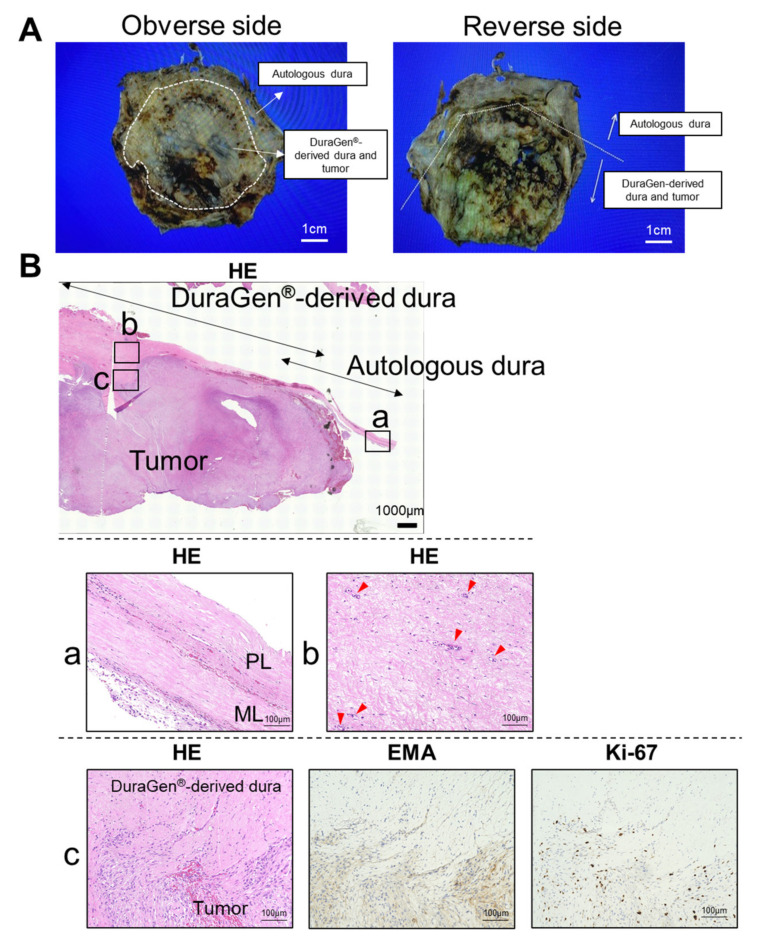
Histopathological finding. (**A**) Macroscopic images of the formalin fixed resected specimen. Left panel: obverse side, right panel: reverse side. (**B**) Coronal section of tumor, autologous dura mater and DuraGen^®^-derived dura mater is shown. The meningeal and periosteal layers are clearly observed in the autologous dura mater (**a**). Collagen fibers running regularly and transversely are observed in the DuraGen^®^-derived dura mater, resembling the meningeal layer of the autologous dura mater (**b**). Neovascularization is also evident in the DuraGen^®^-derived dura mater. Meningioma cell invasion is detected in the DuraGen^®^-derived dura mater (**c**). Immunostainings of EMA and Ki-67 are shown. Serial paraffin sections are immunostained and evaluated for each staining (Original magnification, ×20). Red arrowhead: vascular structure. EMA, epithelial membrane antigen; HE, hematoxylin-eosin stain; ML, meningeal layer; PL, periosteal layer.

## Data Availability

The original contributions presented in the study are included in the article, further inquiries can be directed to the corresponding author.

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
