# Peer review of "Meningioma Cell Invasion into DuraGen-Derived Dura Mater: A Case Report"

_medicines, 2022, doi:10.3390/medicines9040030_

Round 1

Reviewer 1 Report

This is a very well written case report article, however there are some points that could strengthen the evidence presented by the authors and further clarify the messages for the readership of the journal:
1. p.1, line 1: Perhaps it should be stated that it is a case report and not a short communication.
2. p.1, line 12: Local recurrence occurs more or less in most meningioma types, not only in atypical ones. The meningioma in the presented case was initially a WHO I tumor yet it recurred. Perhaps the phrase "Local recurrence frequently occurs in atypical meningioma cases" should be mended.
3. p.3, line 87: EMA staining is not pathognomonic for meningiomas. It may also be observed in metastases and ependymomas. Neither Ki-67 is pathognomonic for meningiomas, as it is a cell proliferation marker, crudely correlating with tumor malignancy. Could you please clarify your observations with regards to these two markers?
4. p.5, lines 128-130: Perhaps the last paragraph of the discussion or a shorter version should be kept for the conclusions section, as a take home message.

Author Response

We are very grateful to the reviewers for their insightful comments and suggestions, which have undoubtedly helped us to improve our manuscript immensely. As indicated in the responses below, we have taken all their comments and suggestions into account when generating the revised version of the manuscript. Responses to the reviewers’ comments appear after the arrows, in blue text.

Reviewer #1:

This is a very well written case report article, however there are some points that could strengthen the evidence presented by the authors and further clarify the messages for the readership of the journal:

  1. p.1, line 1: Perhaps it should be stated that it is a case report and not a short communication.

As the reviewer indicated, we have modified the statement in the revised manuscript.

  1. p.1, line 12: Local recurrence occurs more or less in most meningioma types, not only in atypical ones. The meningioma in the presented case was initially a WHO I tumor yet it recurred. Perhaps the phrase "Local recurrence frequently occurs in atypical meningioma cases" should be mended.

As the reviewer indicated, we have modified the sentences.

Local recurrence may occur in any type of meningioma, but is more likely so in atypical meningiomas.

  1. p.3, line 87: EMA staining is not pathognomonic for meningiomas. It may also be observed in metastases and ependymomas. Neither Ki-67 is pathognomonic for meningiomas, as it is a cell proliferation marker, crudely correlating with tumor malignancy. Could you please clarify your observations with regards to these two markers?

Thank you very much for your comments.

EMA and Ki-67 are used to make a diagnosis and predict the prognosis, respectively. We have added the following sentences in the revised manuscript.

Currently, the most commonly used immunohistochemistry marker for the diagnosis of meningiomas is EMA [Boulagnon-Rombi C. J Neuropathol Exp Neurol. 2017; Schnitt SJ. Am J Surg Pathol. 1986].

The previous study demonstrated that Ki-67 labeling index is the only independent predictor of both tumor recurrence and overall survival in meningioma. It is highly recommended that Ki-67 expression profile is assessed in meningiomas to predict survival [Liu N. Medicine(Baltimore). 2020; Bruna J. Neuropathology. 2007].

References

Boulagnon-Rombi C, Fleury C, Fichel C, Lefour S, Marchal Bressenot A, Gauchotte G. Immunohistochemical Approach to the Differential Diagnosis of Meningiomas and Their Mimics. J Neuropathol Exp Neurol. 2017;76:289-298.

Schnitt SJ, Vogel H. Meningiomas. Diagnostic value of immunoperoxidase staining for epithelial membrane antigen. Am J Surg Pathol. 1986 Sep;10(9):640-9.

Liu N, Song SY, Jiang JB, Wang TJ, Yan CX. The prognostic role of Ki-67/MIB-1 in meningioma: A systematic review with meta-analysis. Medicine (Baltimore). 2020 Feb;99(9):e18644.

Bruna J, Brell M, Ferrer I, Gimenez-Bonafe P, Tortosa A. Ki-67 proliferative index predicts clinical outcome in patients with atypical or anaplastic meningioma. Neuropathology. 2007 Apr;27(2):114-20.

  1. p.5, lines 128-130: Perhaps the last paragraph of the discussion or a shorter version should be kept for the conclusions section, as a take home message.

Thank you very much for your comments. We have modified the section of Conclusion, as the reviewers indicated.

Reviewer 2 Report

This case report entitled, "Meningioma cell invasion into DuraGen-derived dura mater" is reveals that atypical meningioma which frequently recurs may invade and grow into the Dura-Gen material that may be used in post-tumor reconstruction. While this is not surprising, it will be important for neurosurgeons and neuro-oncologists to be made aware of this when considering treatment and post-resection follow-up of high-grade meningiomas.  This case report will be a good contribution to the clinical community. 

Author Response

We are very grateful to the reviewers for their insightful comments and suggestions, which have undoubtedly helped us to improve our manuscript immensely. As indicated in the responses below, we have taken all their comments and suggestions into account when generating the revised version of the manuscript. Responses to the reviewers’ comments appear after the arrows, in blue text.

Reviewer #2:

・This case report entitled, "Meningioma cell invasion into DuraGen-derived dura mater" is reveals that atypical meningioma which frequently recurs may invade and grow into the Dura-Gen material that may be used in post-tumor reconstruction. While this is not surprising, it will be important for neurosurgeons and neuro-oncologists to be made aware of this when considering treatment and post-resection follow-up of high-grade meningiomas.  This case report will be a good contribution to the clinical community.

Thank you very much for your review.

Reviewer 3 Report

The authors deal with an interesting issue as recurrence of intracranial meningiomas, analyzing the role of dura invasion in a case of tumor relapse. In particular, the attention is focused on invasion of a dural patch used for reconstruction during a previous surgery.

This paper is indeed iconographically well-documented; however, many crucial points need to be deeply explored and analyzed.

Please accept the following criticisms:

First of all, since just one case is reported, I suggest modifying the title adding “case report”.

The introduction should be implemented.

In “Case presentation” section, a more detailed histopathological examination report is needed to clearly address a three-time relapse of an apparently benign tumor in an elderly patient (i.e. mitotic index, Ki-67 etc and if modification to this index occurred at recurrence).

How about dural AQP-1 expression? AQP-1 expression is correlated to higher recurrence in meningioma (as in reference n10).

Was MMP-1 analyzed, considering its association to dural invasion? (see ref n.1 below)

Why a Simpson I resection was not attempted/not possible to achieve during the first two surgeries?

Was Duragen placed only during the third surgery? How was the dura reconstructed the first times? Which factors eventually lead to choose a different closure technique in the last procedure?

Conclusion sentences at lines 126-129 (“DuraGen®-derived dura mater may also attract meningioma cell invasions. This indicates that usage of DuraGen® for dural reconstruction with adjacent residual 128 tumor tissue might lead to the extension of meningioma growth”) appear too weakly supported by scientific evidence. Are there specific Duragen features that boost meningioma recurrence more than any other dura patch? Otherwise, should Duragen use forbidden during reconstruction after tumor surgery?

References should be updated.

In conclusion, this paper as it is, seems not amenable to publication, should the authors not provide coherent and convincing arguments to enrich and sustain their manuscript.

Reference:

  1. Murase M, Tamura R, Kuranari Y, Sato M, Ohara K, Morimoto Y, Yoshida K, Toda M. Novel histopathological classification of meningiomas based on dural invasion. J Clin Pathol. 2021 Apr;74(4):238-243. doi: 10.1136/jclinpath-2020-206592. Epub 2020 Jun 16. PMID: 32546547.

Author Response

We are very grateful to the reviewers for their insightful comments and suggestions, which have undoubtedly helped us to improve our manuscript immensely. As indicated in the responses below, we have taken all their comments and suggestions into account when generating the revised version of the manuscript. Responses to the reviewers’ comments appear after the arrows, in blue text.

Reviewer #3:

・The authors deal with an interesting issue as recurrence of intracranial meningiomas, analyzing the role of dura invasion in a case of tumor relapse. In particular, the attention is focused on invasion of a dural patch used for reconstruction during a previous surgery. This paper is indeed iconographically well-documented; however, many crucial points need to be deeply explored and analyzed.

Please accept the following criticisms:

Thank you very much for your review.

・First of all, since just one case is reported, I suggest modifying the title adding “case report”.

We have added “a case report” in the title.

・The introduction should be implemented.

As the reviewer indicated, we have modified the section of Introduction.

・In “Case presentation” section, a more detailed histopathological examination report is needed to clearly address a three-time relapse of an apparently benign tumor in an elderly patient (i.e. mitotic index, Ki-67 etc and if modification to this index occurred at recurrence).

As the reviewer indicated, we have added a more detailed histopathological examination report in the revised manuscript. The Ki-67 labeling index of the 3rd recurrent lesion was higher than those of the previous lesions (initial lesion: 7%, 1st recurrent lesion: 5%, 2nd recurrent lesion: 8%, 3rd recurrent lesion: 32%).

・How about dural AQP-1 expression? AQP-1 expression is correlated to higher recurrence in meningioma (as in reference n10). Was MMP-1 analyzed, considering its association to dural invasion? (see ref n.1 below)

As the reviewer indicated, AQP-1 and MMP-1 are associated with higher recurrence rate and dural invasion. In this case report, we plan to emphasize the existence of meningioma cells in the DuraGen-derived dura mater. The relationship between AQP-1, MMP-1 and meningioma cell invasion in the DuraGen-derived dura mater will be evaluated in the future study using a large number of patients. We have added the limitation in the revised manuscript.

・Why a Simpson I resection was not attempted/not possible to achieve during the first two surgeries?

Thank you very much for your comments.

We decided not to attempt Simpson grade 1 resection while the transverse sinus was patent during the first two surgeries. In contrast, Simpson grade I resection was achieved for the 3rd operation, because the transverse sinus was completely occluded by the tumor.

・Was Duragen placed only during the third surgery? How was the dura reconstructed the first times? Which factors eventually lead to choose a different closure technique in the last procedure?

Duraplasty was performed with a Gore-Tex® dura substitute (W.L. Gore &Associates Inc., Phoenix, AZ, USA) for the 1st and 2nd operations, because, DuraGen has been approved for dural repair since July of 2019 in Japan. Therefore, we could not use DuraGen at the time of 1st and 2nd operations. We have added this information in the revised manuscript.

・Conclusion sentences at lines 126-129 (“DuraGen®-derived dura mater may also attract meningioma cell invasions. This indicates that usage of DuraGen® for dural reconstruction with adjacent residual 128 tumor tissue might lead to the extension of meningioma growth”) appear too weakly supported by scientific evidence. Are there specific Duragen features that boost meningioma recurrence more than any other dura patch? Otherwise, should Duragen use forbidden during reconstruction after tumor surgery?

According to the reviewer’s comment, we have modified the indicated sentences.

Meningioma cells may invade and survive in the DuraGen®-derived dura mater. Whether or not DuraGen® is not appropriate as a dural substitute remains unanswered. Some references supporting this discussion were added in the manuscript.

Future studies analyzing a large number of patients are warranted to validate the findings in this study. We have added the limitation in the revised manuscript.

・References should be updated.

We have updated the references in the revised manuscript.

・In conclusion, this paper as it is, seems not amenable to publication, should the authors not provide coherent and convincing arguments to enrich and sustain their manuscript.

Thank you very much for your review. We have modified this case report, as the reviewers indicated.

・Reference: Murase M, Tamura R, Kuranari Y, Sato M, Ohara K, Morimoto Y, Yoshida K, Toda M. Novel histopathological classification of meningiomas based on dural invasion. J Clin Pathol. 2021 Apr;74(4):238-243. doi: 10.1136/jclinpath-2020-206592. Epub 2020 Jun 16. PMID: 32546547.

Thank you very much for your review. This reference was written by our research group. We have already used this reference in the manuscript.

Round 2

Reviewer 3 Report

I find the authors’ answers satisfactory. I have no further suggestions to improve their manuscript.